# Surface Display—An Alternative to Classic Enzyme Immobilization

**Mateja Lozančić, Amir Sk. Hossain, Vladimir Mrša and Renata Teparić \***

Faculty of Food Technology and Biotechnology, University of Zagreb, Pierottijeva 6, 10000 Zagreb, Croatia
\* Correspondence: rteparic@pbf.hr

**Abstract:** Enzyme immobilization to solid matrices often presents a challenge due to protein conformation sensitivity, desired enzyme purity, and requirements for the particular carrier properties and immobilization technique. Surface display of enzymes at the cell walls of microorganisms presents an alternative that has been the focus of many research groups worldwide in different fields, such as biotechnology, energetics, pharmacology, medicine, and food technology. The range of systems by which a heterologous protein can be displayed at the cell surface allows the appropriate one to be found for almost every case. However, the efficiency of display systems is still quite low. The most frequently used yeast for the surface display of proteins is *Saccharomyces cerevisiae*. However, apart from its many advantages, *Saccharomyces cerevisiae* has some disadvantages, such as low robustness in industrial applications, hyperglycosylation of some heterologous proteins, and relatively low efficiency of surface display. Thus, in the recent years the display systems for alternative yeast hosts with better performances including *Pichia pastoris, Hansenula polymorpha, Blastobotrys adeninivorans, Yarrowia lipolytica, Kluyveromyces marxianus*, and others have been developed. Different strategies of surface display aimed to increase the amount of displayed protein, including new anchoring systems and new yeast hosts are reviewed in this paper.

**Keywords:** surface display; genetic immobilization; yeast cell wall; cell wall proteins

---

## 1. Introduction

Recombinant protein expression and incorporation into the yeast cell wall represents an exceptional tool for engineering and displaying many heterologous proteins. Genes encoding heterologous proteins are commonly fused with the fragments of genes coding for yeast cell wall proteins [1–4]. Furthermore, synthetic protein chimeras that self-assemble into the scaffolds on the yeast surface were developed [5–7], enabling the surface display of protein complexes and enzyme consortia.

The range of systems by which a heterologous protein can be displayed at the cell surface is broad. In general, a good system should employ an anchor protein that would enable efficient transport of the heterologous protein through the secretory pathway, stability, proper folding of the fusion protein, and strong binding at the cell wall surface. Fusion can be done either at the N- or the C-terminal end of the protein, or the protein could be inserted within the sequence of the anchor protein. The location of the fusion might influence folding and post-translational modifications of the protein, its specific activity, stability, and immobilization efficiency. Chemical immobilization of an enzyme to a particular carrier often represents a challenge due to protein conformation sensitivity. Surface anchoring, on the other hand, is a mild and natural way of binding the enzyme to the cell wall in which the yeast cell serves as both a producer and a solid carrier of the recombinant enzyme. Furthermore, reuse of the surface anchored enzyme is enabled by reutilization of the yeast cells [8]. However, it is still challenging to gain the efficiency of display that would be high enough for industrial application. To improve the efficiency and enable surface anchoring of protein complexes, new expression systems and novel techniques for

the display of heterologous proteins have been developed recently. Furthermore, in the recent times, the display systems for alternative yeast hosts that might have better performances than *Saccharomyces cerevisiae* including *Pichia pastoris*, *Hansenula polymorpha*, *Blastobotrys adeninivorans*, *Yarrowia lipolytica*, *Kluyveromyces marxianus*, *Schizosaccharomyces pombe*, *Debaryomyces hansenii*, and others are developing.

## 2. Immobilization Strategies

There are several advantages of immobilization of active proteins on the cell's surface over chemical immobilization of enzymes. Complicated, time-consuming, and expensive isolation and purification of the enzymes, as well as chemical treatment of proteins during immobilization on different matrices, is avoided in cell surface anchoring. Furthermore, while most purified enzymes cannot be reused [9], cells expressing active enzymes on its surface could be used for extended periods without significant loss of enzymatic activity and repeatedly reused [10]. In addition, the processes, including cascade reactions, might be accomplished by using mixtures of strains expressing different enzymes or strains co-expressing more than one enzyme simultaneously.

Surface display systems employ genes encoding heterologous proteins fused with the complete genes or fragments of genes coding for yeast cell wall proteins [1–4]. The yeast cell wall mannoproteins are located in the outer layer of the wall and attached to the inner glucan layer by at least three and most probably four different ways. Some of the proteins are covalently linked to the inner layer, either by ester linkages between specific glutamates and β-1,3-glucan (PIR proteins), or through remnants of glycosylphosphatidylinositol (GPI)-anchors and β-1,6-glucan (GPI proteins), while the others are just adsorbed non-covalently to β-1,3-glucan chains (SCW proteins). Genes coding for recombinant proteins are commonly fused with genes coding for yeast cell wall proteins or their anchoring fragments. According to the characteristics of the protein to be immobilized, fusion can either be done at the N- or at the C-terminal end of the heterologous protein, or it could be inserted inside the sequence of a cell wall protein (Figure 1).

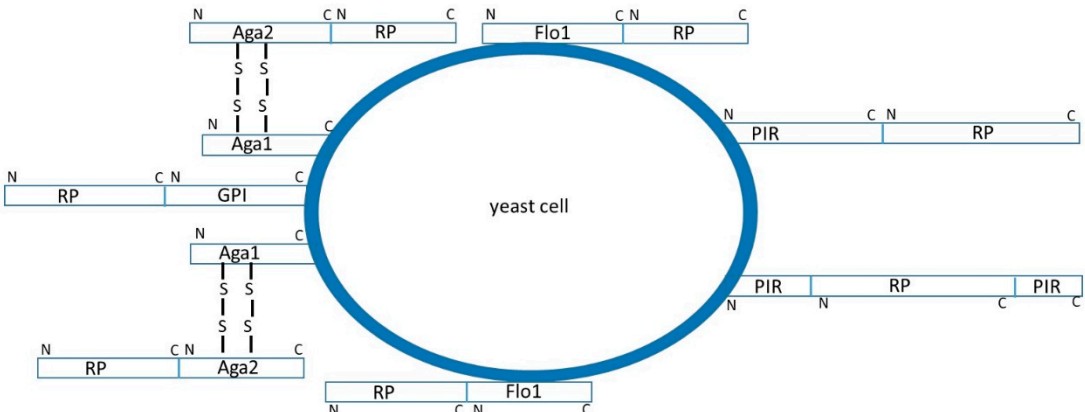

**Figure 1.** Cell surface anchoring systems in yeast: GPI—cell wall protein of GPI group; PIR—cell wall protein of PIR family; RP—recombinant protein to be anchored; Aga1, Aga2, Flo1—individual cell wall proteins.

The most frequently used yeast for surface display is *Saccharomyces cerevisiae* and its cell wall proteins α-agglutinin, a-agglutinin, and Flo1. Flo1 is a lectin-like protein containing the flocculation domain near the N-terminus [11]. Numerous cell-surface display systems in which the N-terminus of the heterologous protein is fused to the Flo1 flocculation domain were constructed. Flo1 system in *S. cerevisiae* was used for surface-display of carboxylesterase EstA [12], α-amylase [13], glucoamylase [14], and few lipases [2,15–17]. Most frequently GPI-anchored proteins used for surface display of heterologous proteins are α-agglutinin [18–24], a-agglutinin [25–30], Cwp2 [4,12,31], and Sed1 [4,32]. Yeast a-agglutinin consists of two subunits, one of which (Aga1) is attached to the cell wall via a GPI anchor while the other (Aga2) is linked to the Aga1 through disulfide bridges. Fusing the

heterologous protein to Aga2 subunit results in its immobilization at the cell surface [25–30]. For a successful N-terminal surface immobilization, a recombinant protein containing the PIR protein and the heterologous protein has to be created [1,33–37]. PIR-proteins' system is particularly suitable for enzymes whose catalytic sites are situated close to their C-termini. Display efficiency of immobilized proteins can be improved by deletion of endogenous *PIR* genes, and in the case of two PIR proteins fused with the enzyme of interest are co-expressed simultaneously in the yeast cell [1]. *PIR3* fusions showed to be more efficient than the *PIR1* and *PIR4* fusions [34].

Although *S. cerevisiae* is still the most frequently used yeast for protein surface display, it has certain shortages, such as recombinant protein hyperglycosylation, poor secretion, and low robustness against temperature and osmotic stress. Thus other yeasts, capable of bearing high osmolarity, temperature fluctuations, and high concentrations of products, are lately more and more explored and used in biotechnology. On the other hand, many attempts to increase protein display efficiency in *S. cerevisiae* have been undertaken including optimization of cell wall anchoring domains, transcription and translation levels of heterologous proteins [38,39], as well as post-translational modifications and transfer in secretory pathway that might also limit successfulness of heterologous protein display.

*2.1. Protein Surface Display in Saccharomyces cerevisiae*

2.1.1. Recombinant Protein Fusion to α-agglutinin

The most frequently used GPI-anchored yeast protein for surface display in *S. cerevisiae* is α-agglutinin. Heterologous protein can either be fused with the entire α-agglutinin, or with a part of it providing the GPI-anchoring signal. Some examples of protein fusions to α-agglutinin are glucoamylase [19], β-galactosidase [18], *Rhizopus oryzae* lipase [20], CM-cellulase [40], xylose isomerase from *Clostridium cellulovorans* [41], laccase from white-rot fungus [22], *Candida albicans* lipase B (CALB) [24], *Rhizomucor miehei* lipase (RML) [23], and β-glucosidase [21,42]. Kaya et al. [42] isolated three new forms of β-glucosidase (Bgl1, Bgl3, and Bgl5) from the filamentous fungi *Aspergillus oryzae* to produce isoflavone aglycones from soybean isoflavones. Isolated enzymes were genetically fused with α-agglutinin and surface displayed in *S. cerevisiae.* All recombinant enzymes efficiently hydrolyzed isoflavone glycosides but showed different substrate specificities. The highest activity was achieved with the recombinant Bgl1. Isoflavone aglycones, produced by the hydrolysis of isoflavone glycosides by β-glucosidase, absorbed readily in the human body and had high bioactivity [43] as structural homologs of human estrogen with a beneficial effect on steroid hormones, prevention of coronary heart disease, and potential prevention of cancer [42,44].

Furthermore, surface displaying yeasts were successfully used as biosensors for bio-adsorption of toxic and rare-metal ions. Metal-binding peptides or proteins immobilized on cell surface enables selective and rapid adsorption of metals and repeated use of yeast cells. Kuroda et al. [45] displayed histidine hexapeptide/α-agglutinin fusion on the surface of *S. cerevisiae* that was capable of adsorption and recovery of copper ions. The aggregation ability of the cells expressing this fusion was limited to appear only in the presence of copper ions by co-expression of copper-responsive transcription factor GTS1 (essential for induction of cell aggregation) under the control of the copper-inducible yeast promoter *CUP1*. The resulting strain both adsorbed copper and self-aggregated, permitting simple removal of cells from the media. The molybdate binding domain of *Escherichia coli* transcription factor ModE has also been displayed on *S. cerevisiae* with the α-agglutinin-anchoring system [46], and further engineered by single amino acid mutation (T163Y) to a selective binder of tungstate [47]. Furthermore, recombinant proteins marked with different peptide tags were expressed on the yeast cell surface and after that used for adhesion of cells to metal oxide surfaces [48]. Surface immobilization of peptide tags might be used for a simple binding of the cells to metal electrodes for the construction of biosensors.

### 2.1.2. Recombinant Protein Fusion to Other GPI-Anchored Proteins

Covalent surface display of heterologous proteins through a-agglutinin binds the protein on the cell surface through disulfide bonding. The flexibility of the Aga2 protein permits both C- or N-terminal fusions of heterologous proteins. This approach was used for surface display of *Trypanosoma cruzi* trans-sialidase [25], *Aspergillus niger* lipase ANL [26], mammalian membrane protein CD47 [27], and a number of mammalian antibodies and receptors [29,49,50]. Yang et al. [51] constructed α-galactosidase fusion to the N-terminus of Aga1 with a flexible linker of 17 amino acids (composed of Ser and Gly) inserted between Aga1 part and α-galactosidase, and accomplished almost doubled display efficiency and 39% activity incensement in comparison to construct in which α-galactosidase was fused to the N-terminus of Aga1 without the linker. Tang et al. [52] used an Aga1 and Aga2 protein pair display system for the construction of synthetic cellulosomes.

The flocculation domain of Flo1, which adheres firmly but non-covalently to the cell wall mannan, is also frequently used for C-terminal immobilization of heterologous proteins [8]. Surface display of glucoamylase [14], *R. oryzae* lipase [2,15], CALB [16], LipB52 from *Pseudomonas fluorescens* B52 [17] and *Burkholderia gladioli* esterase EstA [12] was accomplished through C-terminal fusion with Flo1 flocculation domain.

GPI-anchored yeast cell wall proteins Tip1, Sed1, Cwp1, Cwp2, Ccw12, Spi1, Dan4, Tos6, Srp2, Pry3, and Tir1 have also been used for surface display of recombinant proteins. Some of them, such as Cwp2 and Sed1, were shown to be better carriers than the commonly used α-agglutinin giving six- to eightfold higher levels of displayed heterologous protein at the cell surface [4]. Cwp2 was used as an anchoring protein in the fusion with *Yarrowia lipolytica* lipase Lip2 and with *Burkholderia gladioli* esterase EstA [12,31].

Inokuma et al. [53] showed that the replacement of the anchor domain of α-agglutinin with the anchor domain of Sed1 significantly enhanced the activity of surface-displayed β-glucosidase from *Aspergillus aculeatus* (BGL) and endoglucanase II from *Trichoderma reesei* (EGII). Further optimization of the promoter and signal peptide of Sed1 additionally increased the display efficiency of both recombinant enzymes [53,54]. Inokuma et al. [54] showed that the signal peptide (SP) sequence derived from the *S. cerevisiae SED1* resulted in 1.3- and 1.9-fold higher BGL activity in comparison with BGL activity obtained from the constructs bearing SP from *R. oryzae* glucoamylase (GLUASP) and *S. cerevisiae* α-mating pheromone (MFα1SP), respectively. Using the Sed1 promoter was also favorable because efficient heterologous protein display was achieved by growth fa or prolonged period without any changes in the carbon source. Yang et al. [51] constructed and compared fusions of α-galactosidase with serine- and threonine-rich regions and the GPI binding domains of Sed1, Cwp2, Dan4, Tos6, Srp2, and Pry3. Among them, Dan4 and Sed1 showed the highest display efficiency, but display efficiency and enzyme activities were not entirely dependable. α-Gal-Dan4 had 25% higher activity than that of α-Gal-α-agglutinin, and the others did not show positive effects on enzyme activity. According to that finding, Ser-Gly- linkers were incorporated in the constructs. However, α-Gal-Sed1 showed the greatest improvement (40% higher activity compared to activity without the linker), while the addition of the linker to Cwp2, Tos6, Srp2, and Pry3 enhanced α-Gal activity only by 10% and activity of α-Gal-Dan4 was not increased by the linker addition. Dan4, Aga1, and Sed1 systems were further used to display exoglucanase CBH1 from *Talaromyces emersonii* and β-glucosidase BGL1 from *Saccharomycopsis fibuligera* on the *S. cerevisiae* cell surface. In this experiment, Aga1 and Dan4 showed to be more suitable for displaying large proteins than Sed1 system. Andreu and del Olmo [55] described a new *S. cerevisiae* anchor system through the Spi1 protein, under its own or the *PGK1* promoter, expressed in episomal and centromeric plasmids. The new system was used for the exposure of a number of peptides and proteins of different sizes. *SPI1* is paralog of *SED1*, induced by diauxic shift [56], high osmolarity conditions, basic or acid pH, nitrogen starvation, heat shock, and oxidative stress [57,58]. Its expression in *S. cerevisiae* reaches the maximum between 24 and 36 h of growth [59]. The possibility of displaying certain proteins under exposure to some of those stress conditions might be of great interest in industrial processes.

Genetic immobilizations of *S. cerevisiae* RNase Rny1 and xylose reductase (XR) were achieved by fusions of the Ccw12 signal sequence followed by the catalytic part of the Rny1 or XR, respectively, and the GPI-anchoring signal of the Ccw12 [60,61].

### 2.1.3. Recombinant Protein Fusion to PIR Proteins

Pir1 and Pir2 were fused with three glycosyl transferases (α-1,3-mannosyltransferase, α-1,2-mannosyltransferase, and α-1,2-galactosyltransferase) [33] and *S. cerevisiae* cells expressing these heterologous constructs were shown to conduct sequential synthesis of oligosaccharides identical to those synthesized in vivo.

Pir4 was used for surface display of xylanase A from *Bacillus sp.* BP-7 [35], VP8* fragment of the rotavirus spike protein [36], and two mammalian glycosyl transferases [62]. Catalytic regions of 51 various human fucosyl-, sialyl-, galactosyl-, N-acetylgalactosaminyl-, and N-acetylglucosaminyl transferases were fused with Pir1, Pir3 and/or Pir4 proteins, 40 of which showed activity [37]. Moreover, *PIR1* and *PIR3* fusions showed synergistic effect when expressed simultaneously.

### 2.1.4. Approaches for Increasing Efficiency of Recombinant Protein Surface Display in *S. cerevisiae*

Engineering of Cell Wall Surface and Secretory Mechanisms

Many attempts to increase recombinant protein surface display effectiveness in *S. cerevisiae* have been undertaken. Breinig et al. [12] cloned the *Burkholderia gladioli* EstA gene coding for carboxylesterase between signal peptide of yeast Kre1 and the gene coding for GPI-anchored yeast cell wall proteins Cwp2, or Flo1. Esterase specific activity of Kre1/EstA/Cwp2 showed to be significantly higher than that of Kre1/EstA/Flo1. This result confirmed previous findings that Cwp2 recombinant proteins had higher activities than the activities detected with some other anchoring proteins [4]. Furthermore, they explored whether surface display of esterase would be affected by simultaneous unfolded protein response (UPR) pathway activation. Results showed that the efficacy of cell wall targeting of Kre1/EstA/Flo1 could be further enlarged by co-expression of the Hac1 transcription factor that is a major regulator of the yeast UPR [63]. However, only a slight increase in the Kre1/EstA/Cwp2 cell surface display was observed when Hac1 was overexpressed in high copy numbers. The authors speculated that the reason for the increased anchoring capacity of Cwp2 might be caused by the shorter Cwp2 anchorage sequence (71 amino acids without the signal peptide) which is more suitable to the cellular anchoring apparatus, so its efficacy cannot be additionally improved by UPR activation. On the other hand, the natural length of Flo1 is 1536 amino acids [64], thus using only 104 C-terminal amino acids as the carrier domain might result in a decrease in its anchoring efficacy. However, the cell surface anchoring efficacy of such a shortened domain can be enlarged by increasing the concentration of ER-resident chaperones through Hac1-mediated induction of the UPR. Addition of a 350 amino acid Ser/Thr-rich spacer sequence into the fusions led to a dramatic increase in anchoring efficiency of heterologous proteins [65]. Breinig et al. [12] reported that it was impossible to additionally increase the level of esterase activity by simultaneous expression of the two fusions in a single cell indicating saturation of some critical step during secretion and/or cell wall anchoring. Kuroda et al. [66] accomplished improvement of yeast cell surface display by combining vectors containing *leu2-d* marker (*LEU2* with truncated promoter) and *SED1* disruption in host cells. Fusion of the α-agglutinin-anchoring sequence with red fluorescent protein from *Discosoma sp.* was used as a model protein in this experiment. The use of the *leu2-d* marker is reported to increase the plasmid copy number in yeast resulting in a high expression level even in the presence of leucine [67]. Furthermore, *leu2-d* plasmid showed high stability [66]. Sed1 is a cell wall protein induced by starvation and stress and represents a major cell wall protein in the stationary phase [68]. Accordingly, *SED1* disruption is considered to decrease the competition for cell surface between Sed1 and the α-agglutinin-fused proteins in the stationary phase. A combination of these two approaches gained increased efficiency of surface display. Wentz and Shusta [69] found five display-enhancing genes, three of them coding for cell wall proteins (*SED1*,

*CWP2*, and *CCW12*), one for endoplasmic reticulum-resident protein (*ERO1*), and one for a ribosomal subunit protein (*RPP0*). Products of *CCW12* and *ERO1* showed to be the best enhancers of recombinant protein secretion, and it was previously shown that secretion levels correlate well with the surface display level of recombinant proteins [70]. The screening was done using a plasmid containing a fusion of heterologous protein to the Aga2, and the second one containing a cDNA for an endogenous yeast protein. All selected proteins, except Cwp2, showed maximal secretion levels at 20 °C. Ccw12, Cwp2, Sed1, and Rpp0 assisted secretion of proteins with an originally lower stability and expression rate. However, Ero1 overexpression caused higher secretion for all tested proteins. Rpp0 and Ero1, are known to have a direct role in the synthesis and folding of proteins. Ero1 has a role in bringing oxidizing equivalents to folding disulfide-containing proteins, and it is essential for yeast viability [71]. In the absence of Ero1, reduced proteins accumulate in the ER [72]. The Rpp0 belongs to a large ribosomal subunit [73]. Thus, it may be a limiting constituent in the ribosomal assembly in the case of protein overexpression. Overexpression of the Rpp0 might increase translation capacity. Ccw12, Cwp2, and Sed1 are included in maintaining cell wall stability and in stresses resistance. Therefore, overexpression of these proteins might diminish the stresses caused by heterologous protein secretion. Such results showed that the cell wall components might be involved in secretion improvement.

In spite of all mentioned, many systems still do not show successful protein anchoring with surface accessibility and sufficient display efficiency, although the transcription and translation of heterologous proteins are optimized [38,39], indicating that post-translational modifications and transfer in secretory pathway might also present potential limitations.

Matsuoka et al. [74] reported that the deletion of mannosyl transferase *MNN2* improved the display levels and activities of β-glucosidase from *A. aculeatus* and endoglucanase II from *T. reesei*. The *mnn2* mutant showed decreased level of cell wall mannan, which might result in easier binding of substrates to the active sites of the immobilized enzymes. A similar result was obtained with the *scw4* mutant, deficient in the putative cell wall glucanase Scw4. Such results indicate that changes in the cell wall carbohydrate structure might influence accessibility of the substrates to the heterologous enzymes, as well as the efficiency of surface display.

Tang et al. [75] improved the surface display efficiency of recombinant a-agglutinin fusion with *Clostridium thermocellum* endoglucanase CelA and *S. fibuligera* β-glucosidase BGL1 by engineering vesicle trafficking. Sec12, Sec13, Erv25, and Bos1 proteins, involved in vesicle trafficking from the ER to the Golgi, and Sso1, Snc2, Sec1, Exo70, Sec4, and Ypt32 proteins, involved in vesicle transport from the Golgi to the plasma membrane, were overexpressed in strains expressing CelA/a-agglutinin and BGL1/a-agglutinin. Results showed that overexpression of Sec12, Sec13, and Bos1, as well as overexpression of Snc2, Sec1, Exo70, Sec4, and Ypt32 resulted in increased activity of CelA, while Sso1 did not enhance the display efficiency of CelA. The highest activity improvement was detected in the Bos1 overexpressing strain (71% higher than the control strain). Surface anchoring levels of CelA also increased 83% in Sec12 and 65% in the Bos1 overproducing strain. BGL1 activities were improved in the strains overexpressing Sec12, Sec13, Erv25, Bos1, Snc2, Sec4, and Ypt32 for 24%, 9%, 26%, 29%, 16%, 13%, and 17%, respectively, while overproduction of Sec1 and Exo70 only slightly improved the BGL1 activity. According to that, it can be concluded that engineering both ER to Golgi and Golgi to plasma membrane vesicle trafficking might improve the surface display efficiency and that the vesicle trafficking from ER to Golgi modifications have more impact than those from Golgi to the cell membrane.

Construction of Synthetic Yeast Cellulosomes

In addition to expression of single recombinant proteins at the cell surface, expression systems for the efficient utilization of biomass are also extensively studied. Significant efforts have been invested in the construction of multifunctional minicellulosomes (Figure 2) on *S. cerevisiae* and in engineering of yeast consortia that might carry out biomass hydrolysis and fermentation simultaneously. The challenges in this field are finding appropriate genes for expression in yeast, improving recombinant

enzymes' activities, and maintaining optimal ratios of enzymes in minicellulosomes and yeast strains in consortia.

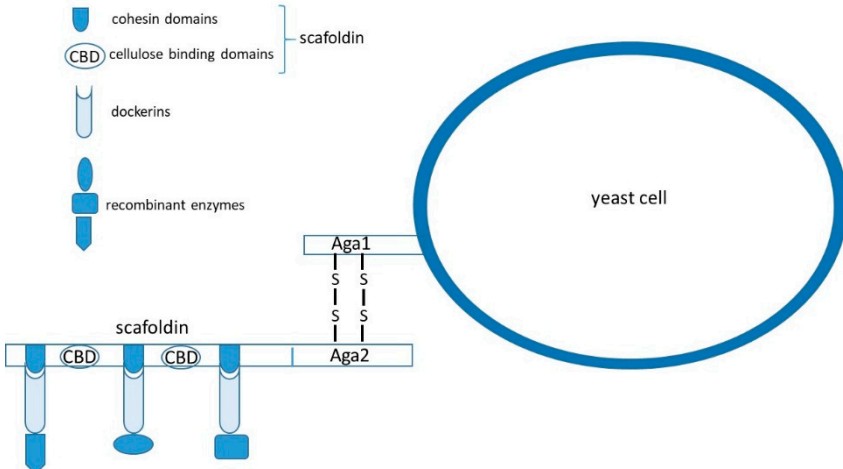

**Figure 2.** Composition of a synthetic cellulosome immobilized on the yeast surface through Aga1/Aga2 anchoring system.

Anaerobic bacteria produce cellulosomes containing scaffoldins made of cellulose-binding domains (CBD) and cohesin domains that can anchor different cellulases containing a dockerin domain in the presence of $Ca^{2+}$ ions [76]. Dockerin domains of the cellulase and cohesin domains of the scaffoldin interact through hydrogen bonding and hydrophobic interactions [77].

Different research groups constructed heterologous cohesin–dockerin pairs from anaerobic bacteria that were used for the self-assembly of cellulosome on the yeast cell surface [6,7]. To improve the efficiency of yeast cellulosomes, Goyal et al. [78] replaced the galactose induced a-agglutinin anchoring system with constitutively expressed α-agglutinin. Other groups tried to increase assembly efficiency of cellulosomes by using two scaffoldins instead of one [79,80]. However, construction of yeast cellulosomes remains challenging, because of the low activity of cellulases, low scaffoldin display level, and the inefficient self-assembly of cellulases on scaffoldins. Ito et al. [5] used the Z domain of protein A as cohesin and the human IgG Fc domain as dockerin, showing that matching protein pairs, other than traditional cellulosomal cohesin–dockerin, might be used for the assembly of cellulosome. Tang et al. [52] used an Aga1 and Aga2 protein pair display system for the construction of synthetic cellulosomes. The Aga1 N-terminal fragment (named tAga1p) was fused with a CBD domain from *Trichoderma reesei* and displayed on the surface of yeast through the C-terminal domain of Aga1 to construct a synthetic scaffoldin (named ScafAGA3) used as the anchor scaffoldin. Aga2 was fused with *C. thermocellum* scaffoldin ScafCipA3, or directly with secreted cellulases, and then assembled into cellulosome through cohesin–dockerin interactions (Figure 2). *T. emersonii* exoglucanase (TedCBH1), *Saccharomycopsis fibuligera* β-glucosidase (Sf-BGL1) and *Clostridium cellulolyticum* endoglucanase (Cc-dCelA) were fused with dockerin from *C. thermocellum*, or with Aga2 for assembly of synthetic cellulosome. In comparison to display level of *C. thermocellum* scaffoldin ScafCipA3, the display level of ScafAGA scaffoldin, comprising repeated tAga1p domains, was considerably enhanced. Binding of cellulases by disulfide bonds also showed to be more effective than that through non-covalent bonding. A two scaffoldin system, comprising ScafAGA as anchor scaffoldin II and ScafCipA3 as scaffoldin I, was also successfully constructed with multiple docking units for heterologous proteins. Although the ethanol production was lower than that of the traditional yeast cellulosome, tAga1p units had a better display efficiency, and the disulfide bonds mediated cellulosome assembly improved the docking efficiency of the cellulases compared with traditional cellulosomes.

## 2.2. Protein Surface Display in Pichia pastoris

*P. pastoris* is the second most widely used yeast for surface display of heterologous proteins. In comparison with *S. cerevisiae*, *P. pastoris* is a better host for the expression of glycoproteins. *S. cerevisiae* is prone to hyperglycosylation that can interfere with displayed protein accessibility, with N-linked mannose chains terminated by α-1,3 attached mannose that is considered to be allergenic in humans. On the other hand, *P. pastoris* has less pronounced glycosylation, and glycoengineered strains of *P. pastoris* providing human-like glycosylation are available. Golgi apparatus of *P. pastoris* is more human-like [81], which might affect the secretion and glycosylation machineries. Furthermore, *S. cerevisiae* is often transformed with episomal vectors. Despite the high transformation efficiencies that are favorable, episomal vectors may be lost under nonselective conditions, so the cells must be cultivated in minimal media. In contrast, integrative vectors are used for transformation of *P. pastoris*, yielding stable transformants even without continuous selective pressure, and they can be cultivated in rich media. Moreover, the availability of vectors containing constitutive GAP or inducible AOX1 promoters offers high flexibility for the expression of recombinant proteins. In comparison to *S. cerevisiae*, this system offers improved secretion efficiency and higher yields, along with accurate glycosylation and folding of recombinant proteins [82,83].

The most frequently used surface display anchors in *P. pastoris* are *S. cerevisiae* proteins Agα1, Aga1, Aga2, Tip1, Sed1, and Flo1, Pir1, and *P. pastoris* Pir1 and Pir2 [84–87]. Wang et al. [84] developed cell-surface display system containing mature peptide of Pir1 from *S. cerevisiae* and the alpha factor secretion signal sequence. Expression of enhanced green fluorescence protein (EGFP) fusion with the N-terminus of the mature peptide of Pir1 (Pir1-a), showed uneven distribution on the *P. pastoris* cell surface. However, the fusion protein EGFP-Pir1-b containing only repetitive sequences of Pir1-a (Pir1-b) was expressed evenly on the *P. pastoris* cell surface. Such results indicate that the repetitive sequences of Pir1 are essential for cell wall binding, while the C-terminal sequence of Pir1 causes the uneven distribution of recombinant proteins in *P. pastoris*. Such results are in good correlation with findings in *S. cerevisiae* [34] indicating that the C- terminal part of Pir1 has a similar function in *P. pastoris* as in *S. cerevisiae*.

Khasa et al. [86] isolated two proteins with internal repeats, PpPir1 and PpPir2, from *P. pastoris*. The *PIR1* and *PIR2* genes' orfs contained 1068 and 972 bases, respectively. Conserved repetitive sequence (SQIGDGQIQATT) was present eight times in the PpPir1 and four times in the PpPir2. The fusion constructs of enhanced green fluorescent protein (EGFP) and PpPir1 or PpPir2 showed correct and uniform localization of EGFP protein on the cell surface of *P. pastoris*.

Thirteen GPI-anchored proteins have been identified in *P. pastoris* by Zhang et al. [88], and three of them (Gcw21, Gcw51, and Gcw61) were used to display CALB. Moura et al. [89] found PpFlo9 flocculin, similar to *S. cerevisiae* Flo1 protein, and used it for surface display of EGFP and CALB. Similar activity and expression of the CALB were accomplished by using proteins PpFlo9 and PpPir1 as anchors [89].

*Kluyveromyces lactis* Yellow Enzyme (KYE) was fused to the C-terminal half of *S. cerevisiae* a-agglutinin and expressed under *AOX1*-promoter by Mergler et al. [90]. The recombinant strain was capable of sorption of the Bisphenol A (BPA) xenoestrogen. Furthermore, Jacobs et al. [91] expressed human erythropoietin (hEPO), mouse interferon beta (mIFN-b), mouse interferon gamma (mIFN-c), and human galectin-1 (hGal-1) on the surface of *P. pastoris* using a-agglutinin-based yeast surface display system.

Shaheen et al. [92] described a method enabling simultaneous secretion and surface display of full-length monoclonal antibody (mAb) molecules in glycoengineered *P. pastoris* producing human-like N-glycans. Fragment (Fc) of the IgG molecule was in this case, fused through a flexible linker to the N-terminus of the *S. cerevisiae* Sed1 and successfully integrated into the cell wall of *P. pastoris.*

Su et al. [85] prepared N-terminal fusion of CALB with α-agglutinin, and C-terminal fusion of CALB with the FS domain of Flo1 and compared them. Somewhat higher expression and activity were detected for the FS-anchored system. Both systems in *P. pastoris* showed better results than those obtained by CALB/α-agglutinin fusion expressed in *S. cerevisiae*. Similar results were obtained by

Jiang et al. [17], who displayed and compared *Pseudomonas fluorescens* B52 lipase LipB52 in *S. cerevisiae* and in *P. pastoris* and observed about fivefold higher enzyme concentration and better stability in *P. pastoris*.

Wasilenko et al. [93] have expressed the hemagglutinin from avian influenza virus (subtype H5N1) fused with C-terminal part of *S. cerevisiae* α-agglutinin on the surface of *P. pastoris*. The recombinant protein retained its natural agglutinating activity and oral vaccination with the yeast expressing it and resulted in the production of virus neutralizing antibodies.

Sun et al. [94] developed a high-efficiency *P. pastoris* display system including two cell wall anchoring genes, the 3′-terminal half Agα1 and Sed1, fused with two genes coding for CALB, separated by a 2A peptide of foot-and-mouth disease virus (FMDV) and combined in a single open reading frame. The fusion of two genes linked by 2A peptide was translated into two independent proteins with high activity and display efficiency.

Surface display of *R. oryzae* lipase in *P. pastoris* using Sed1 as an anchor protein was accomplished by Li et al. [87]. Displayed lipase showed high temperature and pH stability, with the optimum pH of 7.5 and temperature of 40 °C. Furthermore, *S. cerevisiae* Sed1 was used as an anchoring domain for the fragment of antigen-binding (Fab) surface display, expressed in glycoengineered *P. pastoris* with mammalian mannose-type Man5GlcNAc2 N-linked glycans [95].

Gene coding for trehalose synthase (TreS) from *Pseudomonas putida* ATCC 47,054 was fused with the gene coding for Pir1 from *S. cerevisiae* and transformed into *P. pastoris* GS115 [96]. The displayed Pir1-TreS recombinant enzyme was stable over a wide range of pH (6.0–8.5) and temperatures (10–45 °C). The activity of displayed TreS reached approximately equal activity (1108 Ug$^{-1}$) under *PAOX1* and *PGAP* promoter; however, it took a double time span for achieving the same result under the *PAOX1* regulation. Finally, the authors reported that the cell-surface immobilized TreS was capable of producing trehalose from high maltose syrup as substrate at 15 °C and pH 8.0. Furthermore, cells displaying TreS could be recycled three times and still had high catalytic activity.

Jo et al. [97] fused *S. cerevisiae* TIP1 gene or *TIP1* gene fragment encoding the 40 C-terminal amino acids to human lactoferrin cDNA (hLf). Both fusions were expressed in *P. pastoris* SMD 1168. The cells expressing recombinant proteins showed antibacterial activity, confirming that the expressed hLf was biologically active.

### 2.3. Protein Surface Display in Yarrowia lipolytica

*Y. lipolytica* is an obligate aerobe and a non-pathogenic yeast, classified as GRAS (generally recognized as safe), appropriate for cultivation to high cell-density, and capable of forming either yeast cells or hyphae and pseudohyphae depending on the growth conditions. The morphology of *Y. lipolytica* depends mainly on the pH, although cultivation temperature, nitrogen, and carbon sources can also affect the phenotype to some extent [98]. *Y. lipolytica* is an exceptional yeast having the ability to efficiently degrade hydrophobic substrates, such are fats, oils, and n-alkanes [99]. In addition to a number of hexoses (glucose, fructose, mannose), *Y. lipolytica* is capable of using citric, lactic, acetic, propionic, malic, succinic, and oleic acids [100], as well as ethanol at concentrations up to 3%, as sole carbon and energy sources [101]. Processes involving *Y. lipolytica* have been widely used in food industries and in the production of different organic acids, single-cell proteins, enzymes (lipases, esterases, phosphatases, and proteases), and single-cell oils [99,102]. Elaborate genetic tools and heterologous protein expression systems, as well as assembled and annotated genome sequences, made this yeast a promising new host for the expression of heterologous proteins and protein surface display [102]. Moreover, *Y. lipolytica* has certain advantages in comparison to other yeasts, since it secrets proteins by the co-transcription pathway with high secretion capacity (up to 10 gL$^{-1}$) [103] and has a low protein glycosylation level.

*Y. lipolytica* cell wall proteins Ylcwp1 (homologous to *S. cerevisiae* Cwp1), Flo1 (homologous to *S. cerevisiae* Flo1) (YALI0C09031p), YlCwp1, YlCwp2 (YALI0C22836 g), YlCwp3 (YALI0D27214 g),

YlCwp4 (YALI0E11517 g), YlCwp5 (YALI0E31108 g), YlCwp6 (YALI0F18282 g), and YlPir1p were used as anchors for protein surface display in this yeast.

The nucleotide sequence encoding 110 C-terminal amino acids of Ylcwp1 have been used for the assembly of the vector for the cell surface display in *Y. lipolytica* [104]. This system was successfully used to immobilize haemolysin derived from the bacterium *Vibrio harveyi* [104], exoinulinase from *Kluyveromyces marxianus* [105], alkaline protease from the marine yeast *Aureobasidium pullulans* [106], acid protease from *Saccharomycopsis fibuligera* A11 [107], sucrose isomerase from *Pantoea dispersa* [108] and alginate lyase from the marine bacterium *Vibrio sp.* QY101 [109].

Lip2 immobilization on the cell surface of *Y. lipolytica* by the N- and C-domains of the Flo1 homolog (YALI0C09031p) resulted in much higher activity than these previously immobilized through *S. cerevisiae* Cwp2 C-terminal domain [110]. The activity of surface-displayed Lip2 by N- and C-terminal domains of YALI0C09031p reached 9170 and 3200 $Ug^{-1}$ dry cells, respectively. Furthermore, N-terminally fused recombinant lipase was completely linked to the cell surface while C-terminally fused one was partly secreted to the medium (about 30% of the total activity). Yang et al. [111] displayed mannanase (man1) from *Bacillus subtilis* at the surface of *Y. lipolytica* using the C-terminus of Flo1 from *S. cerevisiae* as an anchoring domain.

Yuzbasheva et al. [112] used five putative genes encoding GPI-anchored proteins of *Y. lipolytica* for the cell wall immobilization of *Y. lipolytica* Lip2 protein. Among the C-terminal domains of YlCwp1, YlCwp2 (YALI0C22836 g), YlCwp3 (YALI0D27214 g), YlCwp4 (YALI0E11517 g), YlCwp5 (YALI0E31108 g), and YlCwp6 (YALI0F18282 g) the highest quantity of immobilized Lip2 and lipase activity was obtained with the anchor domains of Ylcwp1 (16,173 ± 1800 $Ug^{-1}$ dry cells), Ylcwp6 (17,700 ± 2101 $Ug^{-1}$ dry cells), and Ylcwp3 (18,785 ± 1130 $Ug^{-1}$ dry cells). These levels of activity are around 70 times higher than the activities of surface-displayed lipases on the *P. pastoris* and *S. cerevisiae* [113–115]. However, a substantial amount of lipase activity was secreted into the medium. Furthermore, immobilized lipase activity was not affected by lyophilization, and it was possible to reuse lyophilized cells repeatedly as whole-cell biocatalysts. The best recombinant enzyme stability was obtained in the fusion of Lip2 with YLcwp3 that after three batches of hydrolysis retained 98% of activity. In contrast, chemically immobilized *Y. lipolytica* lipase on sorbent beads retained only 35% activity after three repeated uses [116].

Finally, N-terminal fusion of Lip2p [117] and xylanase TxXYN from *Thermobacillus xylanilyticus* [118] to the cell wall protein YlPir1p was made. Surface-displayed lipase showed significantly increased thermostability and improved stability in detergents and organic solvents. Despite the high level of immobilized lipase activity, secretion of the enzyme into the medium was detected. *T. xylanilyticus* xylanase TxXYN fused with YlPir1p showed three times higher efficiency (716 $Ug^{-1}$) than the formerly developed YlCwp1 system for the anchoring of TxXYN in *Y. lipolytica*, although 48% of the activity was released in the medium.

### 2.4. Protein Surface Display in Blastobotrys adeninivorans

The first report of *Blastobotrys adeninivorans* is given rather recently by Middelhoven et al. [119], and the species was named *Trichosporon adeninivorans*. In 1991, the species was given the name *Arxula adeninivorans* [120], and it was subsequently renamed to *B. adeninivorans* [121]. All wild-type isolates of this yeast were able to assimilate nitrate and to use soluble starch, melibiose, uric acid, propylamine, butylamine, pentylamine, hexylamine, putrescine, adenine, and guanine as sole sources of carbon, nitrogen, and energy [119]. The *B. adeninivorans* genome is fully sequenced, and the yeast itself is completely characterized. *B. adeninivorans* is capable of growing on media containing up to 20% NaCl [122] and at temperatures up to 48 °C without previous adaptation. However, depending on the temperature, it shows three morphological forms. The cells duplicate by budding up to 42 °C, form pseudomycelia at 42 °C, while above 42 °C they grow in mycelia [123]. After the shift in temperature to below 42 °C, mycelial culture form buds again, showing reversible dimorphism. Wartmann et al. [123] reported that the two different morphological forms show alterations in their secretory properties, but

it is not known if the amount of secreted proteins varies for different proteins [124]. The mechanisms of post-translational modifications in *B. adeninivorans* are not yet well understood, although they seem to depend on the cultivation temperature [125] with O-glycosylation being preferential in budding cells. The genome of *B. adeninivorans* contains four chromosomes with a single rDNA cluster situated close to the Arad1D chromosome's right subtelomere [126]. Linearized cassettes containing a selective marker were created and integrated by homologous recombination in one to three copies into 25S rDNA of *B. adeninivorans*, while transformation with circular plasmids was not successful. In 2004, a *B. adeninivorans* transformation/expression vector based on *E. coli* and *B. adeninivorans* fragments was developed [127]. The vector named Xplor1 contained *B. adeninivorans* 25S rDNA derived sequences permitting specific insertion into the host genome, the *B. adeninivorans TEF1* promoter, the *E. coli hph* gene (conferring resistance) and the *S. cerevisiae PHO5* terminator. This cassette was demonstrated to function in *Debaryomyces hansenii* and *Debaryomyces polymorphus, S. cerevisiae, P. pastoris,* and *Hansenula polymorpha* as well [127]. The Xplor1 was further improved to Xplor®2 system [128], enabling heterologous gene expression either with a yeast rDNA integrative expression cassette (YRC), targeting genes into rDNA clusters, or a yeast integrative expression cassette (YIC), targeting genes randomly into genomic DNA. Both cassettes are involved in the same vector, and the selection of the particular cassette is enabled by two different restriction sites. The Xplor®2 platform, besides YRC and YIC cassettes, contains a multicloning site (MCS) that can accommodate five modules, auxotrophic selection markers, and modules for inducible and constitutive gene expression [127–131].

### 2.5. Protein Surface Display in Schizosaccharomyces pombe

The fission yeast *Schizosaccharomyces pombe* is one of the most frequently used yeast hosts for high-level production of heterologous proteins because it shares numerous biochemical and genetic features with higher eukaryotes [132]. *S. pombe* is evolutionary and taxonomically distant from the budding yeast [133] and has been extensively physiologically and genetically characterized. In terms of cell division control, mRNA splicing and post-translational modifications, *S. pombe* bear a resemblance to a multicellular organism more than budding yeast [134]. Recently, numerous recombinant human proteins and proteins from other sources have been successfully expressed in *S. pombe* [135–137]. However, the first surface display system suitable for *S. pombe* was developed by Tanaka et al. [138]. They demonstrated the display of *Aspergillus aculeatus* beta-glucosidase (BGL) on the surface of *S. pombe* cells using novel anchor proteins of *S. pombe*. The C-terminus of four putative *S. pombe* anchor proteins SPBC359.04c, SPBC947.04, SPBC21D10.06c, and SPBC19C7.05 were fused to beta-glucosidase, and the recombinant protein was expressed in *S. pombe*. The highest enzyme activity (107 U/$10^5$ cells) growth on 2% cellobiose was achieved with SPBC359.04c anchoring protein, followed by SPBC947.04 (44 U/$10^5$ cells) and SPBC21D10.06c (38U/$10^5$ cells).

### 2.6. Protein Surface Display in Debaryomyces hansenii

*Debaryomyces hansenii* is osmotolerant and chemostress tolerant non-pathogenic yeast that can survive in habitats with low water activity (seawater, cheese, fruit, soil) and in high-sugar products [139]. Its osmotolerance and chemostress tolerance permit non-sterile manufacture, cheap waste products as substrates, and high product concentrations, that might be highly useful for biotechnological applications. Furthermore, *D. hansenii* optimal growth temperature is at 20 to 25 °C and it assimilates a broad spectrum of carbon substrates (soluble starch, melibiose, inositol, raffinose) and readily utilizes n-alkanes [140,141]. In accordance with the poor anaerobic growth of *D. hansenii,* its fermentation of sucrose, maltose glucose, trehalose, galactose, and raffinose is weak and lactose fermentation has not been observed. However, it has superior transport capacities, with twice as many carbohydrate and amino acid transporters than *Saccharomyces* species [142]. Terentiev et al. [127] successfully expressed GFP protein in *D. hansenii* using transformation system developed for *B. adeninivorans*, indicating that heterologous genes could be efficiently expressed in *D. hansenii*. This transformation system opens possibilities for heterologous proteins surface display as well.

## 2.7. Protein Surface Display in Kluyveromyces marxianus

The yeast *Kluyveromyces marxianus* has a high growth rate and fermentation at elevated temperature on various low-cost carbon sources [143]. Furthermore, it is regarded as GRAS and is, therefore, suitable for applications in food and pharmaceutics industry. Yanase et al. [144] used this thermotolerant yeast for surface display of β-glucosidase from *Aspergillus aculeatus* and endoglucanase from *Trichoderma reesei* in fusion with the C-terminal half of *S. cerevisiae* α-agglutinin. The recombinant yeast was able to convert cellulosic β-glucan directly to ethanol at 48 °C with the yield of 0.47 g of ethanol per g of degraded β-glucan, matching 92.2% of the theoretical yield.

## 3. Conclusions

In the last two decades, much effort has been focused on the development of molecular tools for surface display in different microorganisms, particularly in yeasts. This is certainly due to the outstanding benefits this methodology offers by simply combining production and secretion of heterologous proteins of biotechnological importance with their immobilization at the cell surface. The method provides simple, fast, and cheap production of large amounts of immobilized proteins and in this respect, is superior to most classic immobilization techniques. The major obstacle to industrial application is a rather low concentration of the immobilized protein in the cell wall. Thus, most attempts aimed to improve the applicability of surface display was directed at increasing in the amount of heterologous protein imbedded at the surface of the cell. These attempts included optimization of the display system, i.e., the cell wall protein or its part used as an anchor, engineering of the cell wall itself by modifying its carbohydrate structure or its protein content and composition, increase in the heterologous protein production, modifications in the secretory mechanism of the host, and the application of different yeasts as inherently better hosts. In many cases, significant improvement of the method has been achieved, but very often, the upgrades of the system were beneficial for the immobilization of some proteins while they showed very limited improvement for the others. Therefore, no universal system, technique, or microorganism has been developed that would fit the requirements of surface immobilization of all heterologous proteins. At the same time, the specific surface activities obtained by surface display still cannot match those obtained by classic immobilization techniques. It is important to note that decades of work on classic immobilization techniques have resulted in numerous different immobilization matrices and binding reactions (for recent reviews see [145–148]) that may have other benefits compared to genetic immobilization. Standard enzyme immobilization may be more expensive, but a proper one can produce biocatalysts with improved stability, activity, selectivity or specificity, improving enzyme purity and even resistance to inhibitors or chemicals [145]. Moreover, mass activity may be very high (e.g., using CLEA, [149]). Using anchored enzymes, some advantages are clear, such as the simplicity of the biocatalyst production process, but some problems may arise, such as the already mentioned moderate volumetric activity, contamination of the medium by other components of the cell, particularly if living cells are used in the catalysis, etc. However, the rate at which this technique has been developing in recent years, and the effort of many research groups, promises further improvements and the eventual commercial industrial application of proteins immobilized at yeast cell walls by surface display methodology.

**Author Contributions:** M.L. and A.S.H. both actively performed a number of experiments in the area of surface display of different enzymes. In frame of their research they have followed the literature in the field, as well. V.M. and R.T. performed a critical analysis of the knowledge obtained so far in this field and composed the text of this review.

**Funding:** This work was supported by a grant provided by the Croatian Science Foundation Nr. HRRZ-IP-2014-09-2837.

**Conflicts of Interest:** The authors declare no conflict of interest.

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
