# Peer review of "Surface Display—An Alternative to Classic Enzyme Immobilization"

_catalysts, doi:10.3390/catal9090728_

Round 1

Reviewer 1 Report

The present paper deals with reviewing the results about "displayed" proteins (enzymes) on the cell wall of microorganisms. The literature background is broad, the scientific description and evaluation of gives a broad panorama of this quickly developing field. Some formal features and the lack of a literature citation should be, however corrected. After these corrections the paper should be accepted for publication.

Deatails. (1)The present Reviewer does not like the denomination of this subject, as "display". It appears, that "grafting, grafted", or "anchoring, anchored" would much better describe the essence of the effects discussed in this paper. The Reviewer knows well, that the term "display, displayed" is broadly used in the relevant literature, but advices tha Authors of this well-done review paper to substitute at least a part of the "display"-s. (2) Reference [52] should be mentioned earlier in the paper, around ref. [8]. (3) The considerations in the paragraph rows 262-268 and row 509 are relevant to the problems of very low or even single molecule concentrations. Two papers dealing with these problems should be cited here: M. Maioli, G. Varadi, R. Kurdi, L. Caglioti, G. Pályi: Limits of the classical concept of concentration. J. Phys. Chem. B 2016, 120, 7438-7445. By the way, the possibility for using the catalytic systems discussed in the present paper for the enantioselective catalytic synthesis of chiral molecules from achiral precursors, is an interesting option. (4) The names of microorganisms are written generally by italics in relevant literature. The present manuscript uses both italics and normal letters, which should be corrected. The Authors are adviced to use italics. (4) In the Abstract and at places where it first appears the complete name of microorganisms should be given, even if the name is well known, e. g. S. cervisiae = Saccharomyces cervisiae and so on. (5) There is a few lines (rows 31-49) in the Introduction typed with bold face letters. This is not "forbidden", but fairly unusual.

Author Response

Reviewer 1. The present paper deals with reviewing the results about "displayed" proteins (enzymes) on the cell wall of microorganisms. The literature background is broad, the scientific description and evaluation of gives a broad panorama of this quickly developing field. Some formal features and the lack of a literature citation should be, however corrected. After these corrections the paper should be accepted for publication.

Deatails. (1)The present Reviewer does not like the denomination of this subject, as "display". It appears, that "grafting, grafted", or "anchoring, anchored" would much better describe the essence of the effects discussed in this paper. The Reviewer knows well, that the term "display, displayed" is broadly used in the relevant literature, but advices tha Authors of this well-done review paper to substitute at least a part of the "display"-s. (2) Reference [52] should be mentioned earlier in the paper, around ref. [8]. (3) The considerations in the paragraph rows 262-268 and row 509 are relevant to the problems of very low or even single molecule concentrations. Two papers dealing with these problems should be cited here: M. Maioli, G. Varadi, R. Kurdi, L. Caglioti, G. Pályi: Limits of the classical concept of concentration. J. Phys. Chem. B 2016, 120, 7438-7445. By the way, the possibility for using the catalytic systems discussed in the present paper for the enantioselective catalytic synthesis of chiral molecules from achiral precursors, is an interesting option. (4) The names of microorganisms are written generally by italics in relevant literature. The present manuscript uses both italics and normal letters, which should be corrected. The Authors are adviced to use italics. (4) In the Abstract and at places where it first appears the complete name of microorganisms should be given, even if the name is well known, e. g. S. cervisiae = Saccharomyces cervisiae and so on. (5) There is a few lines (rows 31-49) in the Introduction typed with bold face letters. This is not "forbidden", but fairly unusual.

Term ˝display, displayed˝ is substituted with ˝anchoring, anchored˝ whenever fitted. Reference 52 is mentioned at position 8. We were not able to correlate the statements from the text with the reference suggested by the reviewer. Perhaps there is a very far (if any) relation but in case we include the reference we should go into more extensive explanations beyond the original scope of this review. So we thought we better not include this reference. All names of microorganisms are corrected to italics and written in complete when first time appears in text. Formatting is corrected to normal.

Reviewer 2 Report

This paper reviews on display systems of proteins on the surface of yeast cells. Various strategies of surface display such as cell wall proteins used for anchor, fusion methods of heterologous proteins to the anchor proteins, and yeast species used for display are well surveyed. Current achievements and limitations of the surface display systems are well summarized. Therefore, this review is considered to be informative for scientists working in the field of enzyme reaction processes. However, there are a few points which would assist comprehension. The comments to this paper are described below:

1. Addition of figures may be helpful for understanding the contents.

Page 2, lines 59-85: localization of cell wall proteins and methods of protein display can be summarized in figures.

Page 6, lines 269-297: cell surface display of cellulosomes is complicated and hard to understand without figures.   

2. Adding subsections in Section 2.1 may be helpful to follow the topics.

For example, “2.1.1 Fusion with agglutinin (p.3, line 96)”, “2.1.2 Other cell wall proteins for surface display (p.3, line 125), “2.1.3 Attempt to increase surface display effectiveness (p.4, line 186”, “2.1.4 Surface display of cellulosomes (p. 6, line 262)”.

3. Page 7, line 334

Is “S. cerevisiae” mistaken for “P. pastoris”?

Author Response

Reviewer 2 This paper reviews on display systems of proteins on the surface of yeast cells. Various strategies of surface display such as cell wall proteins used for anchor, fusion methods of heterologous proteins to the anchor proteins, and yeast species used for display are well surveyed. Current achievements and limitations of the surface display systems are well summarized. Therefore, this review is considered to be informative for scientists working in the field of enzyme reaction processes. However, there are a few points which would assist comprehension. The comments to this paper are described below:

Addition of figures may be helpful for understanding the contents.

Page 2, lines 59-85: localization of cell wall proteins and methods of protein display can be summarized in figures.

Page 6, lines 269-297: cell surface display of cellulosomes is complicated and hard to understand without figures.   

Adding subsections in Section 2.1 may be helpful to follow the topics.

For example, “2.1.1 Fusion with agglutinin (p.3, line 96)”, “2.1.2 Other cell wall proteins for surface display (p.3, line 125), “2.1.3 Attempt to increase surface display effectiveness (p.4, line 186”, “2.1.4 Surface display of cellulosomes (p. 6, line 262)”.

Page 7, line 334

Is “S. cerevisiae” mistaken for “P. pastoris”?

Figures summarizing composition and model of binding of recombinant proteins (Figure 1), and composition of yeast synthetic cellulosome (Figure 2) are added. Subsections are made through the section 2.1. S. cerevisiae was not mistaken for P.pastoris. In the corresponding sentence it was stated that PpFlo9 protein of P. pastoris was similar to the S. cerevisiae Flo1 protein.